# Suicidal ideation and associated factors among pregnant women attending antenatal care at public hospitals of Harari regional state, eastern Ethiopia: A cross-sectional study

**Tilahun Bete** [1]*, **Tilahun Ali**[1], **Tadesse Misgana**[1], **Abraham Negash** [2], **Teklu Abraham**[4], **Dekeba Teshome**[4], **Addisu Sirtsu**[3], **Kabtamu Nigussie**[1], **Abdulkerim Amano**[1]

1 Department of Psychiatry, School of Nursing and Midwifery, College of Health and Medical Sciences, Haramaya University, Harar, Ethiopia, 2 Department of Midwifery, School of Nursing and Midwifery, College of Health and Medical Sciences, Haramaya University, Harar, Ethiopia, 3 Department of Nursing, School of Nursing and Midwifery, College of Health and Medical Sciences, Haramaya University, Harar, Ethiopia, 4 Department of Psychiatry, School Medicine, College of Health and Medical Sciences, Arsi University, Assella, Ethiopia

* tilbete08@gmail.com

**Data Availability Statement:** All relevant data are within the paper and it's supporting information files.

## Abstract

### Background

Suicide is a global issue. It is the third responsible for death among the reproductive age group. Pregnancy is a complicated event and crucial in the life of a woman with considerable physiological, hormonal changes, social, and mental changes. However, third-world countries like Ethiopia have not been investigated well. Therefore, the study planned to assess the magnitude and factors associated with suicidal ideation. Furthermore, it will identify the role of hyperemesis gravidarum on suicidal ideation.

### Method

A Cross-sectional study was employed for 543 pregnant participants attending antenatal care at Hiwot Fana Specialized University Hospital and Jugal General Hospital, Harari regional state, eastern Ethiopia from June 1 to August 1, 2022. The recruited participants were selected by systematic random sampling method. Suicide was assessed using Composite International Diagnostic by interview methods data collection. Epi data and STATA version 14.1 were used for data entry and analysis respectively. Candidate variables were entered into a multivariate logistic regression then those variables that have p-value < 0.05 were considered as significantly associated.

### Results

The magnitude of suicidal ideation in this study was found to be 11.15% at (95% CI: 8.75–14.11). Regarding the associated factor, unwanted pregnancy (AOR = 3.39: at 95% CI = 1.58–7.27), Hyperemesis gravidarum (AOR = 3.65: at 95% CI = 1.81–7.34), having depressive symptoms (AOR = 2.79: at 95% CI = 1.49–5.23), having anxiety symptoms (AOR = 3.37; at 95% CI = 1.69–6.68), experiencing intimate partner violence (AOR = 2.88:

**Funding:** The author(s) received no specific funding for this work.

**Competing interests:** The authors have declared that no competing interests exist.

at 95% CI = 1.11–7.46), and having stress (AOR = 3.46; at 95% CI = 1.75–6.86) were significantly associated variable with suicidal ideation among pregnant women.

## Conclusion and recommendation

This study revealed that suicidal ideation is common among pregnant women. Regarding the associated factors unwanted pregnancy, hyperemesis gravidarum, having depressive and anxiety symptoms, experiencing intimate partner violence, and stress were significantly associated with suicidal ideation. Thus, giving awareness and early screening and interferences for antenatal suicide should be warranted.

## Background

Suicide is a fatal act of terminating one's own life and it is a behavior characterized by thinking or putting personal life at risk of killing oneself [1]. It is a complex process that is classified into three categories: suicidal plan, ideation, and attempt [2]. Suicide ideation (SI) refers to the thought of trying to end one's life with a non-fatal consequence and aiding as the cause of one's death [3] According to a report by WHO, more than 800,000 individuals pass away by suicide, and 20 million attempt suicide annually. It is responsible for 1.4% of deaths, ranking as the 15th top root of death. In sub-Saharan countries, it is underreported. Only 10% of countries report suicide mortality data [4–6].

Pregnancy is a crucial and complicated event in the woman's life with considerable physiological, hormonal changes, social, and mental changes [7, 8]. Hyperemesis gravidarum is not a sign of an underlying psychiatric illness but, it is one of a complex condition that occurs during pregnancy characterized by severe, recurrent episodes of nausea and vomiting developed before the end of the 22$^{nd}$ week of gestational age [9]. Nearly 10% of pregnant mothers with HG continue until the third trimester or throughout the pregnancy [10]. It is a diagnosis of exclusion that is explained by positive ketonuria, persistent nausea, and vomiting, which leads to dehydration, weight loss, and failure to eat and drink normally among pregnant women [11, 12]. The exact cause of HG is not known but some scholars hypothesized that the size increase in placental mass, and hormonal changes during pregnancy [13].

Various studies have reported that hyperemesis gravidarum can associated with suicide. Hyperemesis gravidarum is also one of the predisposing factors for depression, anxiety, and psychological problems [14–16]. These are in turn accountable for more than 90% of suicides [17]. The mechanism of the association between hyperemesis gravidarum and suicide is not fully understood but hypothetically suggested that it is linked with an increment of reproductive hormones like estrogen and progesterone [18, 19] and these hormones are linked to emotional disturbance like depression [20, 21].

Pregnant women suffering from SI. In psychiatry health service around 3–14% are emergency cases [22, 23]. The magnitude of SI in pregnant women is high compared to the general population [24]. Globally suicide is the third leading cause of death among reproductive age particularly among pregnant [25]. It is also one of the reasons for the death of mothers during the perinatal time and a considerable basis of maternal deaths in the perinatal period [26] and one-third of all female patients hospitalized following a suicide attempt [27–29]. In developing countries, it is a main problem for death [8]. The evidence suggested that 1.0–1.7% of pregnancy-related deaths reported in low-income families are indorsed due to suicide [30].

The existing few studies in low and middle-income countries indicate that the magnitude of suicidal ideation ranged between 1.7% -27.5% [24, 28, 31–35]. Being impulsivity in behavior, having a family history of suicide, having a previous diagnosis of psychiatric illness [36, 37], having a comorbid mental illness, particularly diagnosis of depression, anxiety, substance use especially alcohol use [38, 39], unwanted and unplanned pregnancy, induced pregnancy, being unemployment, violence between partner, being young, and having poor social support [40–43] were identified factors for suicidal ideation in both high and low-middle income countries.

The outcome of the study has paramount significance in providing the magnitude of suicidal ideation and associated factors, in identifying the role of hyperemesis gravidarum on suicidal ideation. It is also used as a foundation for concerned bodies who are designed to screen and intervene in suicide since antenatal care is a good chance among pregnant [44–47]. Despite being pregnant being a major risk of having suicidal ideation, there is no study done on the magnitude of the problem in the Eastern part of Ethiopia. The study also identifies the role of hyperemesis gravidarum on suicide in pregnancy. Therefore, the study aimed to figure out the current magnitude of suicidal ideation and its associated factors in the study area.

## Methods

### Study area, design, and period

An institutional-based Cross-sectional study design was conducted. The study was employed in two public hospitals in the Harari region, located 525 KM away from Addis Ababa. In the region, there are 2 public hospitals namely Hiwot Fana Specialized University Hospital and Jugal General Hospital. Hiwot Fana Specialized University Hospital is one of the main academic referral centers in eastern Ethiopia, serving a population of over 5 million people. Currently, the hospital has about 201 beds and 12 case teams to provide referral inpatient and outpatient services to residents of the Harari region and nearby regions. According to hospital records, 10,000 pregnant women attend ANC follow-up each year. Jegula Hospital is the first hospital founded in Harar town. The hospital contains 7 wards and 9 OPD with 347 clinical staff.

The study was carried out from June 1 to August 1, 2022.

### Population

**Source population.** All pregnant women attending antenatal care service in public hospitals of Harari regional state, eastern Ethiopia.

**Study population.** All pregnant women attending antenatal care services in public hospitals of Harari regional state who were available during the study period.

### Eligibility criteria

**Inclusion criteria.** All pregnant women who were attending ANC service at public hospitals during the study were included in this study.

**Exclusion criteria.** Pregnant women who were acutely ill and unable to communicate were excluded from this study. In addition to this, those mothers with hyperemesis gravidarum secondary to other medical conditions such as thyroid diseases and liver diseases were excluded from the study after proven investigation.

## Sample size determination and sampling technique

A single population proportion formula was used to estimate the sample size. The magnitude of suicidal ideation among pregnant women was taken from a previous study which was conducted in southern Ethiopia Jimma, which was 13.3% [33] and by taking a margin of error of 0.03, standard normal distribution $Z_{\alpha/2} = 1.96$, and non-response rate of 10%. Our final sample size was 543.

A systematic sampling technique was used to recruit participants. The average number of pregnant women who attend antenatal care at the public Hospital of Harari regional state is 1224 per month. The sample was proportionally allocated using the monthly average number of overall pregnant women attending ANC from the registration book for each hospital to make it representative. Systematic random sampling was used to select study subjects from each hospital. The interval size (k) is calculated using the following formula. $k = \frac{N}{n} = 2448/543 = 4.5 \approx 4$. So every four persons was selected from each hospital. The first pregnant woman was selected from the first four by lottery method and had to follow up during the data collection period.

## Operational definitions

**Pregnant women**; women who tested positive for HCG test [48] and USG examination [49].

**Hyperemesis gravidarum.** Women who tested positive for ketonuria and had experienced prolonged nausea and vomiting [50].

**Suicidal ideation.** Women who gave "yes" responses to suicidal ideation during the pregnancy period were considered as having suicidal ideation [51].

**Sleep quality.** Women who scored PSQI $\leq 5$ were considered to have good sleep quality and scored$>5$ as poor sleep quality [52].

**Depression.** Participants who scored on DASS-21 "$\geq 10$" are declared as having depressive symptoms [53].

**Anxiety.** Participants who scored on DASS-21 "$\geq 8$" are considered as having anxiety symptoms [53].

**Stress.** Participants who scored on DASS-21 "$\geq 15$" are declared as having stress, [53].

**Intimate partner violence.** Expresses physical violence, sexual violence, and emotional aggression by their intimate partner [54].

**Social support** was measured by using the Oslo social support measuring tool. It is a 3-item questionnaire that was used to assess social support. It has a total score of 3–14 with three categories. Poor "3–8", moderate "9–11", and strong social support "12–14" [55].

Those pregnant women who used at least one of the specified substances (alcohol, khat, tobacco,) for non-medical purposes in the last 3 months were considered positive for current substance use [56].

Gestational age/stage was defined as the pregnancy stage that is categorized as first, second, and third trimester if the duration of pregnancy was 1–3 months, 4–6 months, and 7 months and more, respectively.

## Data collection instruments and procedure

A structured interviewer-administered questionnaire was used and the data was collected through face-to-face interviewing. The questionnaire content includes socio-demographic questions, obstetrics-related conditions, clinical-related factors, psychosocial factors, substance-related factors, psychiatric-related factors, and suicidal ideation questions of the pregnant mothers(S1 Dataset).

The Depression, Anxiety, and Stress Scale-21 (DASS-21) is a set of three self-report scales designed to measure the emotional states of depression, anxiety, and stress. Each of the three DASS-21 scales contains 7 items, divided into subscales with similar content. Recommended cut-off scores for conventional severity labels as normal, moderate, severe, and extremely severe. Each item contributes 0 to 3 points to the sum score, then scores on the DASS-21 were multiplied by 2 to calculate the final score [53]. Various studies demonstrated that the DASS-21 was found to have strong internal consistency with Cronbach's α coefficient ranging from 0.74 to 0.86 for anxiety, 0.77 to 0.92 for depression, and 0.70 to 0.90 for stress [57–59]. In the current study kappa value was 0.91.

Current substance use was assessed by using the adopted version of the WHO (2010) 'Alcohol, Smoking, and Substance Involvement Screening Test (ASSIST), at least one of a specific substance for non-medical purposes within the last 3 months [56]. The average test–retest reliability coefficients (kappas) of ASSIST ranged from a high of 0.90 to a low of 0.58. The average kappas for substance classes ranged from 0.61 for sedatives to 0.78 for opioids. In general, the reliabilities were in the range of good to excellent [60].

Suicidal ideation was assessed using the World Health Organization (WHO) composite international diagnostic interview (CIDI) assessment tool [51]. The tool assesses the suicidal attempt, plan, ideation, and method of attempt. The tool is widely used in Ethiopia among pregnant women and chronic medical illnesses [61–64]. The Amharic version is validated in Ethiopia with percent agreement and kappa ranging from 92.5%-100% and 0.78–1.00, respectively [63–65]. In the current study kappa value was 0.88.

The Pittsburgh Sleep Quality Index (PSQI) was used to assess sleep quality during pregnancy. The tool has 19 questions that are classified into seven categories [66, 67]. Each category has a score of zero to three and the total score ranges from 0–21 [68]. Individuals who score more than five from the total can be considered as poor sleep quality. The tool has a specificity of 86.5% and a sensitivity of 89.6% [52].

Social support was assessed by the Oslo Social Support Scale. The scale has 3 questions that ranged from a minimum score of 3 to a maximum score of 14. Those individuals who scored "3–8" were Poor, "9–11" moderate, and "12–14" were considered as they have strong social support [55]. In the current study, the internal consistency of OSSS-3 was accepted with Cronbach's alpha coefficient of 0.81.

Abuse Assessment Screen (AAS) was used to measure Intimate Partner Violence (IPV). The tool is one of the most valid and widely used IPV screening tools in the pregnant population with a sensitivity of 93–94% and specificity of 55–99%. Pregnant mothers who responded to the questions were considered as having been abused [54].

Laboratory investigations were performed for those participants suspected of underlying medical conditions such as thyroid disease (sent for T3, T4, and TSH) and liver disease (sent for liver function test). Information about Hyperemesis gravidarum and gestational age were taken from the medical chart of the patients as defined as a woman who tested positive for ketonuria and had experienced prolonged nausea and vomiting, and the pregnancy stage that is categorized as first, second, and third trimester, respectively.

## Data quality control

The data was collected by four psychiatric nursing professionals. To ensure the quality of data, training was given to data collectors regarding data collection methods, data collection tools, and how to handle ethical issues. Then the data was collected through face-to-face interviewing of a volunteer pregnant woman attending ANC in public hospitals and questionnaires adopted from WHO that include a tool for diagnostic evaluation of suicidal ideation were used. Each section of the questionnaires was translated from English to local languages, Amharic and Afaan Oromo,

by the language expert and then it was translated back to English for internal consistency. All the materials and equipment were adequately controlled and a pre-test was done on 5% of the total sample size of 27 pregnant women. Regular supervision by the supervisor and principal investigator was made to ensure that all necessary data were appropriately collected.

## Data processing and analysis

The collected data were checked for completeness and consistency, then it was coded, entered into the Epi-Data version 3.1 software, and then exported to the Stata version 14.1 for cleaning and analysis. On bivariate analysis factors with a p-value, less than 0.25 and clinical factors were considered to have an association with the outcome variable. Then multivariate logistic regression analysis was used to identify factors that were independently associated with the outcome variable. The assumption of Multicollinearity was checked by calculating Variance Inflation Factors and there were no problems with multicollinearity identified (no VIF > 10). Additionally, the outlier was checked by calculating residuals and addressed based on the impact of them. The model goodness of fit was checked by Hosmer and Lemeshow test which resulted in 0.68. Then, variables that show statistical significance associated with a p-value less than 0.05 in the multivariate logistic regression analysis were declared to be independent predictors of suicidal ideation.

## Ethics statement

All procedures performed in the study were with the ethical standards of the institutional and/ or national research committee and the Helsinki Declaration of 1964. Before the study began ethical clearance was obtained from the Institutional Health Research Ethics Review Committee (IHRERC) of the College of Health and Medical Sciences of Haramaya University with reference number of IHRERC /069/2022. The college sent a letter of cooperation to public hospitals and a written and signed informed consent was obtained from the head of the institutions before starting the data collection. From all of the participants and their parents/ legal guardians informed, voluntary, written, and signed consent was obtained that declared their agreement to participate in the study. For the minor participants informed, voluntary, written, and signed consent was obtained from their parents/guardians. The information from individual mothers was kept confidential, their identity will not be shown and there will be no dissemination of the information without the respondent's permission. A private room for an interview was prepared; those women who reported suicidal ideation were immediately linked to the psychiatric outpatient department for further evaluation and management. Interviewers were trained to link participants found to be in physically risky conditions and/or in immediate need of counseling to psychologists and psychiatrists. During the data collection COVID-19 prevention protocol was taken action like wearing face masks, maintaining physical distance, and using hand sanitizers, which was being practiced by health professionals in the health care setting as a safety measure.

## Results

### Socio-demographic characteristics of participants

In this study, a total of 538 participants were involved, with an overall 99.07% response rate. The mean age of the participants was 26.53 years with ± 4.48 standard deviation. Of all respondents, 223(41.45%) were between the ages of 25 and 29 years and 314 (58.36%) were Muslim by religion. Regarding occupational status, nearly half of the participants 233 (43.31%) were housewives and 202(37.55%) attended Elementary School. Most of the participant, 462 (85.87%), lives in an urban area (**Table 1**).

**Table 1. Socio-demographic characteristics of pregnant women attending antenatal care at public hospitals in Harari region, eastern Ethiopia, 2022 (N = 538).**

| Variables | Categories | Frequency (N) | Percent (%) |
|---|---|---|---|
| Age | < 20 years | 27 | 5.02 |
| | 20–24 years | 152 | 28.25 |
| | 25–29 years | 223 | 41.43 |
| | > 29 years | 136 | 25.28 |
| Marital status | Married | 519 | 96.47 |
| | Unmarried | 19 | 3.53 |
| Religion | Muslim | 314 | 58.36 |
| | Orthodox | 178 | 33.09 |
| | Protestant | 45 | 8.36 |
| | Catholic | 1 | 0.19 |
| Educational status | Informal education | 51 | 9.48 |
| | Elementary | 202 | 37.55 |
| | Secondary | 152 | 28.25 |
| | Diploma and above | 133 | 24.72 |
| Occupational status | Government employed | 77 | 14.31 |
| | Private | 157 | 29.18 |
| | Farmer | 42 | 7.81 |
| | Student | 29 | 5.39 |
| | Housewife | 233 | 43.31 |
| Residence | Rural | 76 | 14.13 |
| | Urban | 462 | 85.87 |

### Clinically related factors and characteristics of the participants

Of all the participants, more than half of participants 313 (58.18%) were in their third trimester and 293 (54.46%) were multipara. More than one-tenth of participants 62 (11.52%) informed that the present pregnancy was unwanted and around 82 (15.24%) of them had experienced hyperemesis gravidarum. Only three of the respondents had a previous history of psychiatric illness, and one-third of the participants 176 (32.71%) have a family history of mental illness. Around 79 (14.68%) had a family history of suicidal attempts. One-third of them 181(33.64%) had a comorbid chronic medical illness. Of the participants, 138 (25.65%), 100 (18.59%), and 159 (29.55%) had depressive, anxiety, and stress symptoms during their pregnancy respectively (**Table 2**).

### Psychosocial and substance-related factors of the participants

Related to social support, more than one-third 193 (35.87%) of the respondents had strong, 185 (34.39%) had moderate and 160 (29.74%) had poor social support, and nearly one-tenth 48(8.92%) of the defendants experienced IPV. Before three months of this study, 134 (24.91%) of the participants used substances, Of them, 39 (7.25%) chewed khat, 54 (10.04%) drank alcohol, and 22 (4.02%) smoked cigarettes (**Table 3**).

### The magnitude of suicidal ideation among the pregnant women

More than one-tenth of respondents 60 (11.15%) at 95% CI (8.75–14.11) of the pregnant women were reported to have suicide ideation and 20 (3.72%) had suicidal attempts during pregnancy in the past month **(Fig 1)**.

**Table 2. Obstetrics-related characteristics of pregnant women attending antenatal care at public hospitals in Harari region, eastern Ethiopia, 2022 (N = 538).**

| Variables | Categories | Frequency (N) | Percent (%) |
|---|---|---|---|
| Pregnancy by trimester | First trimester | 18 | 3.35 |
| | Second trimester | 207 | 38.48 |
| | Third trimester | 313 | 58.18 |
| Parity | Nullipara | 123 | 22.86 |
| | Primipara | 122 | 22.68 |
| | Multipara | 293 | 54.46 |
| History of abortion | Yes | 67 | 12.45 |
| | No | 471 | 87.55 |
| Abortion intention in the current pregnancy | Yes | 12 | 2.23 |
| | No | 526 | 97.77 |
| Current pregnancy status wanted | Yes | 62 | 11.52 |
| | No | 476 | 88.48 |
| Hyperemesis Gravidraum | Yes | 82 | 15.24 |
| | No | 456 | 84.76 |
| Previous mental illness history | No | 535 | 99.44 |
| | Yes | 3 | 0.56 |
| Family history of mental illness | Yes | 176 | 32.71 |
| | No | 362 | 67.29 |
| Family history of suicidal attempt | Yes | 79 | 14.68 |
| | No | 459 | 85.32 |
| Chronic medical illness | Yes | 181 | 33.64 |
| | No | 357 | 66.36 |
| Depressive symptoms | Yes | 138 | 25.65 |
| | No | 400 | 74.35 |
| Anxiety symptoms | Yes | 100 | 18.59 |
| | No | 438 | 71.41 |
| Stress | Yes | 159 | 29.55 |
| | No | 379 | 70.45 |
| Sleep Quality | Poor | 234 | 43.49 |
| | Good | 304 | 56.51 |

## Factors associated with suicidal ideation among pregnant women

Bi-variable logistic analysis was done to see factors associated with suicidal ideation; Hence, age, unwanted pregnancy, family history of mental illness and suicide, history of chronic medical illness, hyperemesis gravidarum, intimate partner violence, depression, anxiety, stress, social support, and poor sleep quality, and were chosen as candidate variables for multivariate analysis by considering a p-value and clinical factors. Out of those variables treated under multivariate logistic regression analysis, unwanted pregnancy, having depressive, and anxiety symptoms, stress, having intimate partner violence, and hyperemesis gravidarum were significantly associated with suicidal ideation at a p-value <0.05.

Pregnant women who had unwanted pregnancies were 3.39 times [AOR = 3.39(95% CI = 1.58–7.27)] more likely to develop suicidal ideation than those who wanted their current pregnancy. Pregnant women with Hyperemesis gravidarum were about 3.65 times more likely to have suicidal ideation compared to those who had no hyperemesis gravidarum [AOR = 3.65; (95%CI = 1.81–7.34)]. The odds of having suicidal ideation for a pregnant mother with depressive symptoms were 2.79 times more likely than a pregnant mother

**Table 3. Psychosocial and substance-related factors of pregnant women attending antenatal care at public hospitals in Harari region, eastern Ethiopia, 2022 (N = 538).**

| Variables | Categories | Frequency (N) | Percent (%) |
|---|---|---|---|
| Social support | Poor social support | 160 | 29.74 |
| | Moderate social support | 185 | 34.39 |
| | Strong social support | 193 | 35.87 |
| Intimate partner violent | Yes | 48 | 8.92 |
| | No | 490 | 91.08 |
| Overall Current substance user | Yes | 134 | 24.91 |
| | No | 404 | 75.19 |
| Current Khat use | Current Khat use | 39 | 7.25 |
| | Not Current Khat use | 499 | 92.65 |
| Current alcohol use | Current alcohol use | 54 | 10.04 |
| | Not Current alcohol use | 484 | 89.96 |
| Current cigarette use | Current cigarette use | 22 | 4.09 |
| | Not current cigarette use | 516 | 95.91 |

without depressive symptoms (AOR = 2.79; 95% CI = 1.49–5.23). Similarly, pregnant women who had anxiety symptoms were 3.37 times (AOR = 3.37; 95% CI = 1.69–6.68) more likely to develop suicidal ideation than those who had no anxiety symptoms. Pregnant women who experienced intimate partner violence and stress during pregnancy are 3.46 and 2.88 more likely to have suicidal ideation than those who haven't experienced respectively (**Table 4**).

## Discussions

This study is intended to determine the magnitude and associated factors of suicidal ideation among pregnant women at public hospitals in the Harari regional state and it shows the scope of the problem in the study area. The finding of this study revealed that the magnitude of

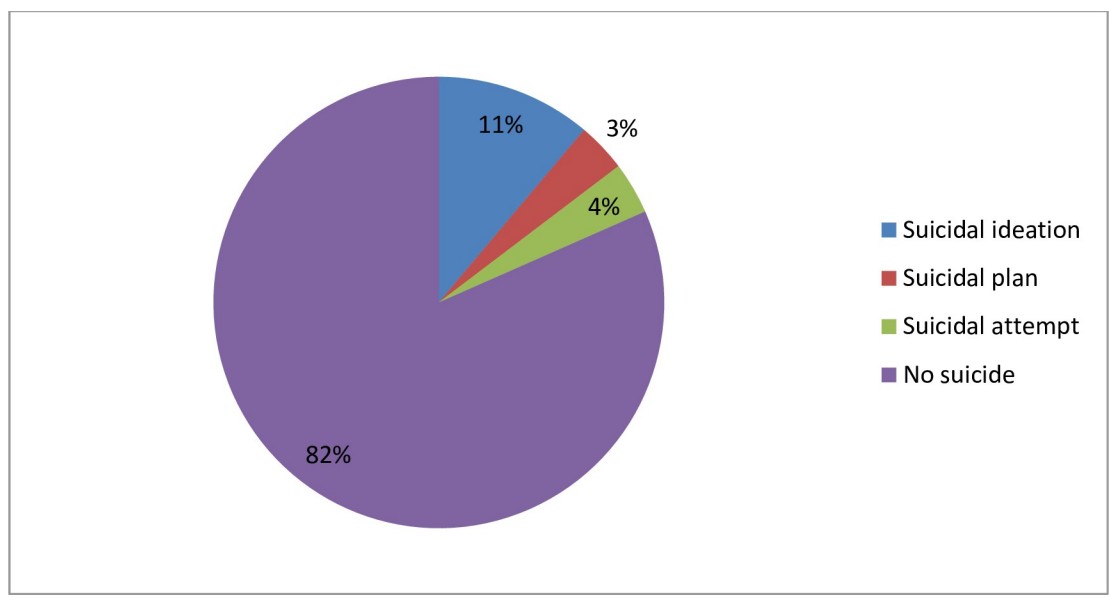

**Fig 1. Distribution of suicidal ideation among pregnant women attending antenatal care at public hospitals in Harari region, eastern Ethiopia, 2022.**

**Table 4. Bivariate and multivariate binary logistic regression of factors associated with suicidal behavior among pregnant women attending antenatal care at public hospitals in Harari region, eastern Ethiopia, 2022 (N = 538).**

| Variables | Category | Suicidal Ideation | | COR (95% CI) | AOR (95% CI) | P-values |
|---|---|---|---|---|---|---|
| | | Yes | No | | | |
| Age | < 20 | 3 | 24 | 1.18(0.31–4.47) | 0.63(0.13–3.08) | 0.663 |
| | 20–24 | 17 | 135 | 1.19(0.56–2.55) | 0.88(0.18–4.15) | 0.867 |
| | 25–29 | 27 | 196 | 1.13(0.65–2.62) | 0.54(0.11–2.71) | 0.450 |
| | >29 | 13 | 123 | 1 | 1 | 1 |
| Current pregnancy wanted | unwanted | 16 | 46 | 3.42(1.79–6.53) | 3.39(1.58–7.27) | **0.002** |
| | Wanted | 44 | 432 | 1 | 1 | 1 |
| Family history of mental illness | Yes | 23 | 153 | 1.32(.76–2.29) | 1.33(0.56–3.13) | 0.517 |
| | No | 37 | 325 | 1 | 1 | 1 |
| Family history of suicide | Yes | 6 | 73 | 0.62(0.26–1.49) | 0.55(0.21–1.45) | 0.225 |
| | No | 54 | 405 | 1 | 1 | 1 |
| History of chronic medical illness | Yes | 21 | 160 | 1.07(0.61–1.88) | 1.06(0.48–2.37) | 0.879 |
| | No | 39 | 318 | 1 | 1 | 1 |
| HyperemsisGravidam | Yes | 23 | 59 | 4.41(2.45–7.94) | 3.65(1.81–7.34) | **<0.001** |
| | No | 37 | 419 | 1 | 1 | 1 |
| Social support | Poor | 24 | 136 | 1.83(.94–3.54) | 1.53 (0.71–3.29) | 0.278 |
| | Moderate | 19 | 166 | 1.18(0.59–2.36) | 1.02(0.44–2.37) | 0.955 |
| | Strong | 17 | 176 | 1 | 1 | 1 |
| Depression | Yes | 34 | 104 | 4.70(2.69–8.19) | 2.79(1.49–5.23) | **0.001** |
| | No | 26 | 374 | 1 | 1 | 1 |
| Anxiety | Yes | 24 | 76 | 3.53(1.99–6.25) | 3.37(1.69–6.68) | **<0.001** |
| | No | 36 | 402 | 1 | 1 | 1 |
| Stress | Yes | 30 | 129 | 2.70(1.57–4.66) | 3.46(1.75–6.86) | **<0.001** |
| | No | 30 | 349 | 1 | 1 | 1 |
| Sleep quality | Poor | 34 | 200 | 1.82(1.06–3.13) | 1.26(0.66–2.39) | 0.479 |
| | Good | 26 | 278 | 1 | 1 | 1 |
| Intimate partner violence | Yes | 11 | 37 | 2.68(1.28–5.58) | 2.88(1.11–7.46) | **0.029** |
| | No | 49 | 471 | 1 | 1 | 1 |

*COR- Crudes odds ratio, AOD- Adjusted odds ratio, CI- Confidence Interval, 1 –reference

suicidal ideation found to be 11.11% (95% CI: 8.75–14.11). Regarding the associated factors unwanted pregnancy, hyperemesis gravidarum, experiencing IPV, having depressive and anxiety symptoms, and stress were identified variables that have significant association with suicidal ideation among pregnant women.

The finding of this study (11.11%) is in line with the studies conducted in Southern Ethiopia Gedeo zone 9.3% [69], Southwest Ethiopia Jimma 13.3% [33], Pakistani 11.8% [70], and Peru 8.8% [71]. But this finding is higher than the studies conducted in India 6% [72], and Brazil 6.3% [73]. The variation might be related to the tool difference used to assess suicidal ideation, the participants' period of pregnancy (trimester), and socioeconomic status variation. This study used CIDI whereas the previous studies used the, revised suicidal behavior questionnaire (SBQ-R) [72], and Self-Report Questionnaire-20 [73], and all of them have different accuracy levels in assessing suicide ideation. For instance, the study conducted in Brazil used the suicidal part of SRQ-20 which included nine questions, and of this suicidal behavior was evaluated by one question that may lead to a decrease the attention and concentration for the question. The other reason for the variation might be the difference in

the inclusion criteria of participants. The Indian study includes pregnant women with a gestational age between five weeks to 20 weeks and excludes those mothers who had a previous history of mental illness and who are using psychoactive substances use which may decrease the prevalence, while this study includes all pregnant women except acutely ill [24, 72, 74]. On top of that, socioeconomic status variation may additionally contribute to the difference in the prevalence of suicide ideation.

On the opposite side, the finding is lower than from studies done in Brazil which was a cross-sectional study that reported the magnitude of suicidal ideation 23.53% [75], the epidemiological study conducted in Egypt reported that the magnitude of SI 20.4% [76], and Peru's study 16.8% [77]. The discrepancy might be due to the tool used to assess SI. This study used CIDI while previous studies used the Beck Depression Inventory (BDI), Primary Care Evaluation of Mental Disorders, and Patient Health Questionnaire (PHQ-9) which measure suicide ideation in 1 week, 30 days, and 15 days, respectively. Another reason might be the difference in inclusion and exclusion criteria. The previous studies exclude pregnant mothers who have a previous history of psychiatric disorder or substance use, and they include pregnant mothers with only who are married and with a gestational age of between 20–40 weeks, while the current study included all pregnant mothers. Another reason might be the socio-demographic and cultural differences and religious roles in which they give meaning to suicide. For example in Ethiopia, the religious leader teaches suicide is forbidden and it is considered as sin. Additionally, most communities in developing countries including Ethiopia consider reporting suicide as not a culturally acceptable and stigmatizing issue, so this could be a possible reason for the low prevalence of suicidal behavior in our study compared to studies done in other countries.

Regarding the associated factors, the odds of having suicidal ideation is 3.65 more likely to occur among pregnant mothers who are diagnosed with hyperemesis gravidarum than those mothers who aren't diagnosed with hyperemesis gravidarum. This is supported by studies done in the United Kingdom [14], Canada [15], and Egypt [78]. Hyperemesis gravidarum disrupts individuals and family life, causing impairment in social and occupational functioning and loss of job. This in turn may cause thoughts of suicide. In another way, Hyperemesis gravidarum, as a physical stressor, may predispose individuals to depression and anxiety [79, 80], again the anxiety and depression may lead individuals to the thought of suicide [81].

This study pointed out that the odds of having suicidal ideation were 6.3 times more likely to occur among unwanted pregnancies compared to wanting one. This finding agrees with previous studies in Brazil [73] and Ethiopia [34]. This might be because the women with an unwanted pregnancy may not be ready to take the responsibility to give birth and to handle the child and they may be dependent economically on their family so due to this reason they may be stressed and think of ending their life. Another justification might be if the sexual partner does not accept, and support the pregnancy it is stress for the pregnant mother that leads them to end their life. In addition to this pregnancy that happened due to sudden forced sexual abuse may result in a crisis for mothers even wanting to end their lives [34].

The finding of this study shows that depressive symptoms were identified as a predicted variable for suicidal ideation for pregnant women. The finding agrees with the findings of Ethiopia Jimma [33], Brazil [73, 75], India [72], and Egypt [76]. The main reason for this could be, that depression is one of the main responsible factors for suicide. Mothers who have depression have hopelessness, decreased need for interest, and internal feelings of sadness, and guilt which lead them to think of ending their lives. Another justification might be the depletion of serotonin and epinephrine in depression has a direct association with suicidal

thoughts through loss of concentration and impairment of judgment, since suicide is a good judgment for poor decision-making individuals [71].

This study suggested that pregnant mothers who are living with anxiety symptoms have more suicidal ideation than those who have no anxiety symptoms. The finding is similar to the previous studies of South Africa [24], Egypt [76], Pakistani [70], and Brazil [82] Pregnancy is a dual life and a very sensitive period. At this time pregnant women become stressed, anxious, and worried about their pregnancy, their child, losing their body shape, taking responsibility, and worrying about the complications of pregnancy in their lives. So all these factors are a potential cause of suicide [75].

Another finding from this study is the odds of having suicidal ideation during pregnancy were 2.43 times occur among mothers who had experienced IPV than those who didn't. The result agrees with the previous studies of Brazil [73, 75], India [72] South Africa [24], and Pakistani [70]. The possible implication for this might be IPV could be the threats of an action that can physical, sexual, verbal, psychological, and emotional violence that can occur either at a private level or at a public leads them to suicide [83, 84]. The finding of the study also showed that women who have stress were 2.88 times more likely to have suicidal ideation as compared to those women who didn't have stress. This is similar to the study done in Ethiopia, Jimma [33]. The implication for this might be the hormonal changes during pregnancy may cause changes in behavior, in mood, and may fear and worry for the future [33, 43, 85].

## Strengths and limitations of the study

The strength of the study was including a relatively large sample size and used standardized tools. This study has some limitations. Due to the nature of a cross-sectional study design, we could not explore the cause-and-effect relationships between suicide and the independent variables. Second, a face-to-face interview method might induce recall bias and social desirability response bias. This was attempted to be reduced by conducting an anonymous survey, ensuring confidentiality, use of self-administered for those literate, and framing questions in a non-threatening and non-judgmental manner.

## Conclusion and recommendation

This study revealed that suicidal ideation among pregnant women was found to be a common problem. Regarding the associated factors unwanted pregnancy, hyperemesis gravidarum, experiencing IPV, having depressive and anxiety symptoms, and stress were identified variables that are found to be significantly associated with suicidal ideation among pregnant women. This suggests the need to strengthen the awareness of suicidal behaviors and the need to evaluate the effectiveness of the national health strategy in addressing suicidal behaviors among pregnant women. Additionally, providing integrated mental health services during ANC is needed to reduce coexisting mental disorders including suicide during pregnancy. Interventions targeted among pregnant women who had a previous history of hyperemesis gravidarum, symptoms of common mental disorders, experiencing IPV, and women with unplanned pregnancies should be warranted.

## Supporting information

**S1 Dataset. The data set was used to determine suicidal ideation and associated factors among pregnant women in the eastern part of Ethiopia.**
(DOCX)

**S1 File.**
(SAV)

## Acknowledgments

We would like to acknowledge Haramaya University for providing us with ethical clearance. In addition, we extend our thanks to our participants, supervisor, and data collectors.

## Author Contributions

**Conceptualization:** Tilahun Bete, Tilahun Ali, Tadesse Misgana, Dekeba Teshome, Addisu Sirtsu, Kabtamu Nigussie, Abdulkerim Amano.

**Data curation:** Tilahun Bete, Teklu Abraham, Kabtamu Nigussie.

**Formal analysis:** Tilahun Bete, Tilahun Ali, Teklu Abraham, Kabtamu Nigussie, Abdulkerim Amano.

**Funding acquisition:** Tilahun Bete, Abraham Negash, Teklu Abraham, Dekeba Teshome, Addisu Sirtsu, Kabtamu Nigussie.

**Investigation:** Tilahun Bete, Abraham Negash, Dekeba Teshome, Addisu Sirtsu, Kabtamu Nigussie.

**Methodology:** Tilahun Bete, Tilahun Ali, Tadesse Misgana, Abraham Negash, Dekeba Teshome, Kabtamu Nigussie, Abdulkerim Amano.

**Project administration:** Tilahun Bete, Abraham Negash, Teklu Abraham, Dekeba Teshome.

**Resources:** Tilahun Bete, Abraham Negash, Dekeba Teshome.

**Software:** Tilahun Bete, Tadesse Misgana, Dekeba Teshome, Abdulkerim Amano.

**Supervision:** Tilahun Bete, Tilahun Ali, Tadesse Misgana, Dekeba Teshome, Abdulkerim Amano.

**Validation:** Tilahun Bete, Abdulkerim Amano.

**Visualization:** Tilahun Bete, Tadesse Misgana, Teklu Abraham, Abdulkerim Amano.

**Writing – original draft:** Tilahun Bete, Tilahun Ali, Tadesse Misgana, Abraham Negash, Teklu Abraham, Dekeba Teshome, Addisu Sirtsu, Kabtamu Nigussie, Abdulkerim Amano.

**Writing – review & editing:** Tilahun Bete, Tilahun Ali, Tadesse Misgana, Abraham Negash, Teklu Abraham, Dekeba Teshome, Addisu Sirtsu, Kabtamu Nigussie, Abdulkerim Amano.

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
