## [Decision Letter · Decision Letter 0]

2 Jan 2024

PONE-D-23-31139Suicidal ideation and associated factor among pregnant women in Eastern part of Ethiopia.PLOS ONE

Dear Dr. Gebremariam,

Thank you for submitting your manuscript to PLOS ONE. After careful consideration, we feel that it has merit but does not fully meet PLOS ONE’s publication criteria as it currently stands. Therefore, we invite you to submit a revised version of the manuscript that addresses the points raised during the review process.

We look forward to receiving your revised manuscript.

Kind regards,

Chalachew Kassaw Demoze, Msc

Academic Editor

PLOS ONE

“The whole required (material and humanitarian) cost for this research work was covered by Haramaya University.”

3. In the online submission form, you indicated that [The datasets used and/or analyzed during the current study are available from the corresponding author upon reasonable request.].

Additional Editor Comments:

Dear editor , I would like to thank you for writing a good paper.

But I do have some concerns and corrections

1. Have you done any labratory experiments? for a statement, " those mothers with hyperemesis gravidarum secondary to other medical conditions such as thyroid

diseases and liver diseases were excluded from the study after proven investigation." If so, please state the data collection tool.

2. Please rewrite a reliability description (specificity, sensitivity) for each tool you have used to assess the outcome and associated factors.(World Health Organization (WHO) composite international diagnostic

interview (CIDI) assessment tool, Alcohol, Smoking, and Substance Involvement Screening Test (ASSIST), Depression, Anxiety, and Stress Scale - 21 (DASS-21) , Abuse Assessment Scale (AAS) and Oslo Social support Scale.

3. It is better to describe some of the clinical variables ( Hypermesisgravidrum and gestational age information) that were taken from their medical card or chart)

4. it is better to correct spelling errors ( at the regression table; medical was written as medical) and number alignment ( center , right and left , look at the regression table)

5. Have you made statistical assumptions? If so, please describe: symptoms of suicidal ideation is also found in depressive symptoms of DASS.) Do you solve this collinearity? , state in detail on data analysis

6. On your exclusion criteria, you did not mention any exclusion of those who have a previous history of mental illness. This might affect your outcome variable if you include how do you treat your data ( there are 3 in your data )

7. It is better to rewrite the title of the place of study. I mean, it misled a reader as it was conducted in all public hospitals found in eastern Ethiopia.

8. In the discussion , it is better to explain a difference other than data collection tool difference.

Reviewers' comments:

Reviewer's Responses to Questions

**Comments to the Author**

1. Is the manuscript technically sound, and do the data support the conclusions?

Reviewer #1: Yes

Reviewer #2: Yes

2. Has the statistical analysis been performed appropriately and rigorously? 

Reviewer #1: Yes

Reviewer #2: Yes

3. Have the authors made all data underlying the findings in their manuscript fully available?

Reviewer #1: Yes

Reviewer #2: Yes

4. Is the manuscript presented in an intelligible fashion and written in standard English?

Reviewer #1: Yes

Reviewer #2: Yes

5. Review Comments to the Author

Reviewer #1: Dear authors,

Thank you for your contribution to the field. I have had the opportunity to review your manuscript and found it well-written. However, there are some points you need to address to improve the quality of your work, and I have mentioned them section by section as follows:

Title and Abstract

1. The title is interesting and it addresses the current public health problem, but you should rewrite it as “Suicidal ideation and its associated factor among pregnant women in the eastern part of Ethiopia: A cross-sectional study.”

2. In the methods section of the abstract, you should incorporate the specific study area and study period.

3.In the result section of the abstract, paraphrase the following sentence to make it easier for the reader: Experiencing intimate partner violence and stress (AOR= 3.46; at 95% CI= = 1.75–6.66) and (AOR = 2.88; at 95% CI = 1.11–7.36) were significantly associated with suicidal ideation among pregnant women. “

4. Put keywords in ascending order

introduction  

5. Overall, the introduction is well-written and adheres to the systematic writing of the background (deductive approach), and it maintains coherence between paragraphs. However, try to improve the grammatical errors through proofreading.

Methods  

The methods are clear, and the statistical approaches used in this study are appropriate. I have only three suggestions to improve the paper:

6. The sample size calculation formula does not need to be included in the calculation. The author just needs to convey what formula is used along with the margin of error set to get the required sample size.

7. In lines 102-107, you should specify your study area. In which hospital did you conduct your study? Let us know more about your study hospitals.

8. In lines 120–122, why do you exclude those with hyperemesis gravidarum secondary to other medical conditions such as thyroid diseases and liver diseases after proven investigation? It needs further justification if thyroid diseases and liver diseases have a direct linkage with suicidal ideation; if it is so, don’t forget to cite your reference for your justification.

9. In line 127, why do you use a margin of error of 0.03 instead of 0.05?

10.Line 135-140 Why do you take average monthly attendants since your study period is 2 months? You calculated the k value in the wrong way. Please readjust it.

11. In Line 141, why do you select from the first three since your k value is 2?

12. In lines 143–146,  you should provide the questionnaire that was used for your study in a supplementary file and cite it in the method section as a supplemental file.

13. In Line 166 (IPV), you should provide both full words and abbreviations when you use it for the first time.

14. In lines 168–169, move the sentence and merge it under data quality control.

15. In data quality control, did you provide training for data collectors? If so, you should state it in your manuscript.

16. In Line 170-189, move your operational definitions above the data collection instruments and procedures.

17. Cite reference for the operational definitions of pregnant women, hyperemesis gravidarum, Suicidal ideation and sleep quality

Results

The results are relevant, but you should revise them based on the following comment to improve the manuscript.

18.Line 214, “{26.53” should be replaced by "26.53 years.”

19. Line 215, do you think the word most is suitable for 58.36%?

20. Line 217, delete “with the largest proportion."

21. In Table 1, you should re-categorize the age by using the standard age category. If you have a study participant whose age is less than 18, you should specify it by adding the category <18 years, since it will provide us with good results.

22. It is difficult to assess the average monthly income for your study participants since either they are not civil servants or unless you did a wealth index.

23. Please try to change Table 4 into a pie chart format

24. Lines 260–262, rewrite each factor separately: “ The odds ratio of having suicidal ideation for a pregnant mother with depressive and anxiety symptoms were 2.79 and 3.37 compared to their counterparts (AOR = 2.79; (95% CI= = 1.49–5.23) and AOR = 3.37; (95% CI = 1.69–6.68), respectively.”

25. In Table 5, age less than 20, how do you run the regression for cell value less than 5, which is "3"? It is a critical issue and must be addressed by re-categorizing the age value.

26. In the Table 5 suicidal ideation column, put the suicidal value of yes before no for a better understanding

Discussion  

The discussions are explained well enough and based on the results. But to improve it, you should revise it based on the following comments:

27. In Line 282, try to justify how the tool difference is the possible reason for the variation. Try to justify by raising ideas about each tool and their possibility to increase or decrease their estimation of prevalence.  Please try to think beyond tool difference and inclusion criteria

28. Lines 290–292, try to justify how the tool difference is the possible reason for the variation. Try to justify by raising ideas about each tool and their possibility to increase or decrease their estimation of prevalence. Please try to think beyond tool difference and inclusion criteria

29. Line 301, you should specify by stating the specific study area for “This is supported by studies (13–15).”

30. Line 301-303 is not directly linked to your justification. You should better delete it. “A cross-sectional study conducted in the United Kingdom reported that 52.1 percent of participants thought to terminate their pregnancy, and 4.9 percent of them terminated their pregnancy owing to hyperemesis gravidarum.”

Strengths and Limitations of the study

31. Line 344, please clarify this sentence:evidence-based laboratory tests Which lab? To assess what?

32. In lines 344–345, it is good to state recall bias and social desirability bias as limitations, but you should better state your effort to reduce those biases in this section.

33.Line 344–346 is a non-sense paragraph; you are expected to paraphrase it “However, recall bias, social desirability, a cross-sectional study design that cannot allow establishing a temporal relationship between suicidal ideation, and significant associated factors were the limitations of the study.

Conclusion

34. Your conclusion is aligned with the implication of your study rather than the mere figure of the results and is also drawn from your main finding. I am okay with that.

35. In the recommendation section, why do you only stick with hyperemesis gravidarum since you have many findings?

Reviewer #2: 1.It will be more helpful to the readers if the setting where the study was conducted is mentioned in the abstract part.

2.It also makes the research more influential if the research owner gives full feedback on the findings. For example, unwanted pregnancy is a factor for SI, but there is nothing to what should be done

3.It would not be good to use one or two of the references that are out of date

4.The Overview of suicidal ideation among the study population is not organized well based on Global to local contexts .

5.author text document spacing should be double spacing according to the journal guide line

6. PLOS authors have the option to publish the peer review history of their article (what does this mean?). If published, this will include your full peer review and any attached files.

Reviewer #1: **Yes: **Eyob Ketema Bogale

Reviewer #2: No

---

## [Author Response · Author response to Decision Letter 0]

13 Feb 2024

REBUTTAL LETTER

We were pleased to have an opportunity to revise our manuscript entitled “Suicidal ideation and associated factor among pregnant women in Eastern part of Ethiopia”. In the revised manuscript, we have carefully considered journal requirements, the editor's and reviewers suggestions and comments and we have tried to address it accordingly. The editor’s and reviewer’s comments were very helpful overall, and we are appreciative of such constructive feedback on our original submission. After addressing the issues raised, we feel the quality of the paper is much improved. 

Sincerely,

On behalf of all authors, 

Tilahun Bete

Author’s response: We have checked the templates and made the adjustments to meet the journal requirements. Some of them are:

 We corrected all major sections (Abstract, Introduction, Materials and Methods, Results, Discussion) to level 1 heading, bold type, 18pt font, and sentence case 

 We corrected sub-sections of major sections to Level 2 heading, bold type, 16pt font, and sentence case. 

 We corrected sub-sections of within level 2 headings to level 3 heading, bold type, 14pt font, and sentence case. 

 We used appropriate file naming.

We hope that it now fits the style requirements. Thank you

“The whole required (material and humanitarian) cost for this research work was covered by Haramaya University.”

Authors’ response: We have accepted the comment and we removed any funding-related text from the acknowledgments section or other areas of the manuscript in the revised version. We do not want to change the funding statement stated on the online submission form since it is similar with what we have stated in the funding section. Thank you.

3. In the online submission form, you indicated that [The datasets used and/or analyzed during the current study are available from the corresponding author upon reasonable request.]

Authors’ response: We have accepted the comment and we have uploaded all the data underlying the findings described in the manuscript as supplementary information. Then, we have changed the statement in Availability of data and materials section as “All relevant data are within the paper and it’s supporting information files” in the revised version of the manuscript. Thank you. 

Authors’ response: We have accepted the comment and we moved it to the Methods section and deleted it from any other section. Thank you. 

Additional Editor Comments:

Dear Author, I would like to thank you for writing a good paper.

But I do have some concerns and corrections

1. Have you done any labratory experiments? for a statement, " those mothers with hyperemesis gravidarum secondary to other medical conditions such as thyroid

diseases and liver diseases were excluded from the study after proven investigation." If so, please state the data collection tool.

Authors’ response: Dear editor, thank you for your concern. In order to decrease the potential confounders, we did the lab investigations for those participants suspected for underlying medical conditions like thyroid disease (sent for T3, T4, and TSH) and liver disease (sent for liver function test like SGOT, SGPT). We included this in the data collection tool section in the revised manuscript. Thank you. 

2. Please rewrite a reliability description (specificity, sensitivity) for each tool you have used to assess the outcome and associated factors.(World Health Organization (WHO) composite international diagnostic

interview (CIDI) assessment tool, Alcohol, Smoking, and Substance Involvement Screening Test (ASSIST), Depression, Anxiety, and Stress Scale - 21 (DASS-21) , Abuse Assessment Scale (AAS) and Oslo Social support Scale.

Authors’ response: We have accepted the comment and we have tried to rewrite a reliability description (specificity, sensitivity) for each tools used in this study in the revised version of the manuscript. Thank you.

3. It is better to describe some of the clinical variables ( Hypermesis gravidrum and gestational age information) that were taken from their medical card or chart)

Authors’ response: Dear editor, thank you for the comment. We addressed the comment and described the variables that were taken from the medical charts at the end of data collection instruments section in the revised manuscript. Thank you. 

4. it is better to correct spelling errors ( at the regression table; medical was written as medical) and number alignment ( center , right and left , look at the regression table)

Authors’ response: Dear editor, we have accepted the comment and corrected the spelling errors and number alignment. Thank you. 

5. Have you made statistical assumptions? If so, please describe: symptoms of suicidal ideation is also found in depressive symptoms of DASS.) Do you solve this collinearity? , state in detail on data analysis

Authors’ response: Dear editor, thank you for the comment. We did the multi-collinearity and extreme outlier assumption checking by calculating Variance Inflation Factors (VIF) and residual respectively, including for suicidal component in the DASS with suicidal ideation. We have described and included this in the revised version of the manuscript. However, we didn’t check linearity assumption since no continuous independent variable was used in the data. Thank you. 

6. On your exclusion criteria, you did not mention any exclusion of those who have a previous history of mental illness. This might affect your outcome variable if you include how do you treat your data ( there are 3 in your data )

Authors’ response: We have included those participants who have a previous of mental illness because they had minor mental health problem which is less likely to experience suicidal behavior and consequently less likely to impact the outcome variable. Therefore, we included them in order to avoid deliberate exclusion of the participants. Thank you. 

7. It is better to rewrite the title of the place of study. I mean, it misled a reader as it was conducted in all public hospitals found in eastern Ethiopia.

Authors’ response: Dear editor, we have accepted the comment and rewrite the title as “Suicidal ideation and associated factors among pregnant women attending antenatal care at public hospitals of Harari regional state, eastern Ethiopia” as suggested. Thank you 

8. In the discussion , it is better to explain a difference other than data collection tool difference.

Authors’ response: We have accepted the comment and we tried to describe the possible explanations for the discrepancy occurred between the current study and previous studies other than tool difference in the revised manuscript. Thank you. 

Reviewers' comments:

Reviewer #1: Dear authors,

Thank you for your contribution to the field. I have had the opportunity to review your manuscript and found it well-written. However, there are some points you need to address to improve the quality of your work, and I have mentioned them section by section as follows:

Title and Abstract

1. The title is interesting and it addresses the current public health problem, but you should rewrite it as “Suicidal ideation and its associated factor among pregnant women in the eastern part of Ethiopia: A cross-sectional study.”

Authors’ response: Dear reviewer, we have accepted the comment and rewrite the title as “Suicidal ideation and associated factors among pregnant women attending antenatal care at public hospitals of Harari regional state, eastern Ethiopia: A cross-sectional study” as suggested by you and editor. Thank you. 

2. In the methods section of the abstract, you should incorporate the specific study area and study period.

Authors’ response: We have accepted the comment and we included a specific study area and study period. Thank you. 

3.In the result section of the abstract, paraphrase the following sentence to make it easier for the reader: Experiencing intimate partner violence and stress (AOR= 3.46; at 95% CI= = 1.75–6.66) and (AOR = 2.88; at 95% CI = 1.11–7.36) were significantly associated with suicidal ideation among pregnant women. “

Authors’ response: We have accepted the comment and paraphrase it accordingly in the revised manuscript. Thank you. 

4. Put keywords in ascending order

Authors’ response: We have accepted the comment and we put the keywords in ascending order. Thank you. 

introduction 

5. Overall, the introduction is well-written and adheres to the systematic writing of the background (deductive approach), and it maintains coherence between paragraphs. However, try to improve the grammatical errors through proofreading.

Authors’ response: Dear reviewer, thank you for the comment. We have tried to improve the grammatical error in the revised version of the manuscript. 

Methods 

The methods are clear, and the statistical approaches used in this study are appropriate. I have only three suggestions to improve the paper:

6. The sample size calculation formula does not need to be included in the calculation. The author just needs to convey what formula is used along with the margin of error set to get the required sample size.

Authors’ response: We have accepted the comment and removed the formula as suggested. Thank you. 

7. In lines 102-107, you should specify your study area. In which hospital did you conduct your study? Let us know more about your study hospitals.

Authors’ response: We have accepted the comment and specified the study area including the two public hospitals where the study was done. Thank you. 

8. In lines 120–122, why do you exclude those with hyperemesis gravidarum secondary to other medical conditions such as thyroid diseases and liver diseases after proven investigation? It needs further justification if thyroid diseases and liver diseases have a direct linkage with suicidal ideation; if it is so, don’t forget to cite your reference for your justification.

Authors’ response: Thank you for the concern. One of the primary reasons for excluding individuals with hyperemesis gravidarum secondary to other medical conditions from the study was the complexity and heterogeneity of their medical condition. Thyroid diseases and liver diseases encompass a wide range of disorders, each with its own unique pathophysiology, clinical manifestations, and treatment approaches. Including individuals with hyperemesis gravidarum secondary to these diverse conditions in a study may introduce significant variability that could confound the results and make it challenging to draw meaningful conclusions. That is why we prefer to exclude them. Thank you. 

9. In line 127, why do you use a margin of error of 0.03 instead of 0.05?

Authors’ response: We took proportion (P) of 13.3% from the previous study conducted in Jimma as a magnitude of suicidal ideation among pregnant women, which was resulting small sample size. Therefore, in order to increase the sample size, the margin error of 3% was taken. Thank you. 

10.Line 135-140 Why do you take average monthly attendants since your study period is 2 months? You calculated the k value in the wrong way. Please readjust it. 

Authors’ response: In order to get the average flow of attendants, we dividing the annual total number into months. However, we made an error on the calculating the K value, we supposed to take the two month period, and we readjusted it in the revised manuscript as “The interval size (k) is calculated using the following formula. k=N/n =2448/543 = 4.5 ≈4.So every four persons was selected from each hospital”. Thank you for your constructive comment. 

11. In Line 141, why do you select from the first three since your k value is 2?

Authors’ response: We took the comment and corrected it in the revised version of the manuscript as “The first pregnant woman was selected from the first four by lottery method”. Thank you. 

12. In lines 143–146, you should provide the questionnaire that was used for your study in a supplementary file and cite it in the method section as a supplemental file.

Authors’ response: We have accepted the comment and we have uploaded all the data underlying the findings described in the manuscript as supplementary information and we have cited it as supplemental file in data collection tool sub-section in the methods section of the revised the manuscript. Thank you. 

13. In Line 166 (IPV), you should provide both full words and abbreviations when you use it for the first time.

Authors’ response: We have accepted and addressed the comment. Thank you. 

14. In lines 168–169, move the sentence and merge it under data quality control.

Authors’ response: We have accepted and addressed the comment. Thank you.

15. In data quality control, did you provide training for data collectors? If so, you should state it in your manuscript.

Authors’ response: Yes we provide the training for data collectors regarding data collection methods, data collection tools, and how to handle ethical issues. We have stated this in the revised manuscript. Thank you. 

16. In Line 170-189, move your operational definitions above the data collection instruments and procedures.

Authors’ response: We have accepted and addressed the comment. Thank you. 

17. Cite reference for the operational definitions of pregnant women, hyperemesis gravidarum, Suicidal ideation and sleep quality

Authors’ response: We have accepted and addressed the comment. Thank you.

Results

The results are relevant, but you should revise them based on the following comment to improve the manuscript.

18.Line 214, “{26.53” should be replaced by "26.53 years.”

19. Line 215, do you think the word most is suitable for 58.36%?

20. Line 217, delete “with the largest proportion."

Authors’ response: We have accepted and addressed the comments from 18 to 20. Thank you.

21. In Table 1, you should re-categorize the age by using the standard age category. If you have a study participant whose age is less than 18, you should specify it by adding the category <18 years, since it will provide us with good results.

Authors’ response: Thank you for your concern. We have got only two participants who aged than 18 years and this small number makes us unable to compare them with other age category. That is why we have merged it as less than 20 years. 

22. It is difficult to assess the average monthly income for your study participants since either they are not civil servants or unless you did a wealth index.

Authors’ response: We agree with your concern. Since we didn’t do the wealth index and less reliable to put it in income, we omit it from the revised manuscript. Thank you. 

23. Please try to change Table 4 into a pie chart format

Authors’ response: We have accepted the comment and changed table 4 into pie chart. Thank you. 

24. Lines 260–262, rewrite each factor separately: “ The odds ratio of having suicidal ideation for a pregnant mother with depressive and anxiety symptoms were 2.79 and 3.37 compared to their counterparts (AOR = 2.79; (95% CI= = 1.49–5.23) and AOR = 3.37; (95% CI = 1.69–6.68), respectively.”

Authors’ response: We have accepted the comment and we rewrite it as suggested. Thank you. 

25. In Table 5, age less than 20, how do you run the regression for cell value less than 5, which is "3"? It is a critical issue and must be addressed by re-categorizing the age value.

Authors’ response: Thank you for your concern. The regression can run the cell value if it is not 0, but the total frequency of the specific category should be more than 5, which is 27 in this case. That is why we categorize it as mentioned in the manuscript. Thank you. 

26. In the Table 5 suicidal ideation column, put the suicidal value of yes before no for a better understanding

Authors’ response: We have accepted and addressed the comment as suggested. Thank you.

Discussion 

The discussions are explained well enough and based on the results. But to improve it, you should revise it based on the following comments:

27. In Line 282, try to justify how the tool difference is the possible reason for the variation. Try to justify by raising ideas about each tool and their possibility to increase or decrease their estimation of prevalence. Please try to think beyond tool difference and inclusion criteria

28. Lines 290–292, try to justify how the tool difference is the possible reason for the variation. Try to justify by raising ideas about each tool and their possibility to increase or decrease their estimation of prevalence. Please try to think beyond tool difference and inclusion criteria

Authors’ response: We have accepted the comment and tried to describe the possible explanations for the discrepancy occurred between the current study and previous studies other than tool difference in the revised manuscript. Additionally, we also elaborated how the tools difference affects the outcome (suicide). Thank you.

29. Line 301, you should specify by stating the specific study area for “This is supported by studies (13–15).”

Authors’ response: We have accepted and addressed the comment as suggested. Thank you.

30. Line 301-303 is not directly linked to your justification. You should better delete it. “A cross-sectional study conducted in the United Kingdom reported that 52.1 percent of participants thought to terminate their pregnancy, and 4.9 percent of them terminated their pregnancy owing to hyperemesis gravidarum.”

Authors’ response: We have accepted the comment and deleted the stated statement since it is not a relevant justification. Thank you.

Strengths and Limitations of the study

31. Line 344, please clarify this sentence:evidence-based laboratory tests Which lab? To assess what?

32. In lines 344–345, it is good to state recall bias and social desirability bias as limitations, but you should better state your effort to reduce those biases in this section.

33.Line 344–346 is a non-sense paragraph; you are expected to paraphrase it “However, recall bias, social desirability, a cross-sectional study design that cannot allow establishing a temporal relationship between suicidal ideation, and significant associated factors were the limitations of the study.

Authors’ response: For question 31 to 33, we have accepted the comments and we extensively rewrite the strength and limitations section in the revised version of the manuscript. 

Conclusion

34. Your conclusion is aligned with the implication of your study rather than the mere figure of the results and is also drawn from your main finding. I am okay with that.

35. In the recommendation section, why do you only stick with hyperemesis gravidarum since you have many findings?

Authors’ response: We have accepted the comment and we added the recommendations for the remaining main findings. Thank you.

Reviewer #2: 1.It will be more helpful to the readers if the setting where the study was conducted is mentioned in the abstract part.

Authors’ response: We have accepted the comment and we included a specific setting were the study was conducted in the revised version of the manuscript. Thank you. 

2.It also makes the research more influential if the research owner gives full feedback on the findings. For example, unwanted pregnancy is a factor for SI, but there is nothing to what should be done

Authors’ response: We have accepted the comment and we added the recommendations for the all main findings as suggested. Thank you.

3.It would not be good to use one or two of the references that are out of date

Authors’ response: We have accepted the comment and we have tried to use latest references. Thank you. 

4. The Overview of suicidal ideation among the study population is not organized well based on Global to local contexts. .

Authors’ response: We have accepted the comment and re-arranged the introduction part as suggested. Thank you. 

5.author text document spacing should be double spacing according to the journal guide line

Authors’ response: We have checked the journal guideline and made the adjustments to meet the journal requirements. Thank you. 

---

## [Editor Report · Decision Letter 1]

15 Feb 2024

PONE-D-23-31139R1Suicidal ideation and associated factors among pregnant women attending antenatal care at public hospitals of Harari regional state, Eastern Ethiopia: A cross-sectional studyPLOS ONE

Dear Dr. Gebremariam,

Thank you for submitting your manuscript to PLOS ONE. After careful consideration, we feel that it has merit but does not fully meet PLOS ONE’s publication criteria as it currently stands. Therefore, we invite you to submit a revised version of the manuscript that addresses the points raised during the review process.

We look forward to receiving your revised manuscript.

Kind regards,

Chalachew Kassaw Demoze, Msc

Academic Editor

PLOS ONE

Journal Requirements:

Additional Editor Comments:

PLOS one Reviewer comment (Suicidal ideation and associated factors among pregnant women attending antenatal care at public hospitals of Harari regional state, Eastern Ethiopia: A cross-sectional study).

1. On the introduction section

1.1. it is better to change the second reference 1 ( you cited two times )

2. Discussion

2.1. It is better to revise line number 315-325 , it is better to cite for each measurement of suicidal ideation e.g. EPDS ( cite here a study ) , (SBQ-R) ( cite here), ……..

2.2. It is better to revise line number 330-341, it is better to add citation previous studies ( cite here) , Ethiopia (cite here , in study) and studies done in other countries ( cite here).

2.3. It is better to revise line number 375-378 , It is not advisable to compare with USA studies , please find studies done in low income countries , Asian …..

Finally it is better to revise spelling, grammar and vocabulary corrections on the whole section of a manuscript.

---

## [Author Response · Author response to Decision Letter 1]

22 Feb 2024

REBUTTAL LETTER

We were pleased to have an opportunity to revise our manuscript entitled “Suicidal ideation and associated factor among pregnant women in Eastern part of Ethiopia”. In the revised manuscript, we have carefully considered journal requirements, the editor's and reviewers suggestions and comments and we have tried to address them accordingly. The editor’s and reviewer’s comments were very helpful overall, and we are appreciative of such constructive feedback on our original submission. After addressing the issues raised, we feel the quality of the paper is much improved. The second comment of the editor is responded to at the end of the first comment of the editor.

Sincerely,

On behalf of all authors, 

Tilahun Bete

Author’s response: We have checked the templates and made the adjustments to meet the journal requirements. Some of them are:

 We corrected all major sections (Abstract, Introduction, Materials and Methods, Results, Discussion) to level 1 heading, bold type, 18pt font, and sentence case 

 We corrected sub-sections of major sections to Level 2 heading, bold type, 16pt font, and sentence case. 

 We corrected sub-sections of within level 2 headings to level 3 heading, bold type, 14pt font, and sentence case. 

 We used appropriate file naming.

We hope that it now fits the style requirements. Thank you

“The whole required (material and humanitarian) cost for this research work was covered by Haramaya University.”

Authors’ response: We have accepted the comment and we removed any funding-related text from the acknowledgments section or other areas of the manuscript in the revised version. We do not want to change the funding statement stated on the online submission form since it is similar with what we have stated in the funding section. Thank you.

3. In the online submission form, you indicated that [The datasets used and/or analyzed during the current study are available from the corresponding author upon reasonable request.]

Authors’ response: We have accepted the comment and we have uploaded all the data underlying the findings described in the manuscript as supplementary information. Then, we have changed the statement in Availability of data and materials section as “All relevant data are within the paper and it’s supporting information files” in the revised version of the manuscript. Thank you. 

Authors’ response: We have accepted the comment and we moved it to the Methods section and deleted it from any other section. Thank you. 

Additional Editor Comments:

Dear editor , I would like to thank you for writing a good paper.

But I do have some concerns and corrections

1. Have you done any laboratory experiments? for a statement, " those mothers with hyperemesis gravidarum secondary to other medical conditions such as thyroid

diseases and liver diseases were excluded from the study after proven investigation." If so, please state the data collection tool.

Authors’ response: Dear editor, thank you for your concern. In order to decrease the potential confounders, we did the lab investigations for those participants suspected for underlying medical conditions like thyroid disease (sent for T3, T4, and TSH) and liver disease (sent for liver function test like SGOT, SGPT). We included this in the data collection tool section in the revised manuscript. Thank you. 

2. Please rewrite a reliability description (specificity, sensitivity) for each tool you have used to assess the outcome and associated factors.(World Health Organization (WHO) composite international diagnostic

interview (CIDI) assessment tool, Alcohol, Smoking, and Substance Involvement Screening Test (ASSIST), Depression, Anxiety, and Stress Scale - 21 (DASS-21) , Abuse Assessment Scale (AAS) and Oslo Social support Scale.

Authors’ response: We have accepted the comment and we have tried to rewrite a reliability description (specificity, sensitivity) for each tools used in this study in the revised version of the manuscript. Thank you.

3. It is better to describe some of the clinical variables ( Hypermesis gravidrum and gestational age information) that were taken from their medical card or chart)

Authors’ response: Dear editor, thank you for the comment. We addressed the comment and described the variables that were taken from the medical charts at the end of data collection instruments section in the revised manuscript. Thank you. 

4. it is better to correct spelling errors ( at the regression table; medical was written as medical) and number alignment ( center , right and left , look at the regression table)

Authors’ response: Dear editor, we have accepted the comment and corrected the spelling errors and number alignment. Thank you. 

5. Have you made statistical assumptions? If so, please describe: symptoms of suicidal ideation is also found in depressive symptoms of DASS.) Do you solve this collinearity? , state in detail on data analysis

Authors’ response: Dear editor, thank you for the comment. We did the multi-collinearity and extreme outlier assumption checking by calculating Variance Inflation Factors (VIF) and residual respectively, including for suicidal component in the DASS with suicidal ideation. We have described and included this in the revised version of the manuscript. However, we didn’t check linearity assumption since no continuous independent variable was used in the data. Thank you. 

6. On your exclusion criteria, you did not mention any exclusion of those who have a previous history of mental illness. This might affect your outcome variable if you include how do you treat your data ( there are 3 in your data )

Authors’ response: We have included those participants who have a previous of mental illness because they had minor mental health problem which is less likely to experience suicidal behavior and consequently less likely to impact the outcome variable. Therefore, we included them in order to avoid deliberate exclusion of the participants. Thank you. 

7. It is better to rewrite the title of the place of study. I mean, it misled a reader as it was conducted in all public hospitals found in eastern Ethiopia.

Authors’ response: Dear editor, we have accepted the comment and rewrite the title as “Suicidal ideation and associated factors among pregnant women attending antenatal care at public hospitals of Harari regional state, eastern Ethiopia” as suggested. Thank you 

8. In the discussion , it is better to explain a difference other than data collection tool difference.

Authors’ response: We have accepted the comment and we tried to describe the possible explanations for the discrepancy occurred between the current study and previous studies other than tool difference in the revised manuscript. Thank you. 

On the introduction section

1.1. it is better to change the second reference 1 ( you cited two times )

Authors’ response: We have accepted the comment and we changed it accordingly. 

2. Discussion

2.1. It is better to revise line number 315-325 , it is better to cite for each measurement of suicidal ideation e.g. EPDS ( cite here a study ) , (SBQ-R) ( cite here), ……..

Authors’ response: We have accepted the comment and we cited the reference for each tool separately. You can see the revise manuscript. 

2.2. It is better to revise line number 330-341, it is better to add citation previous studies ( cite here) , Ethiopia (cite here , in study) and studies done in other countries ( cite here).

We have accepted the comment and we cited accordingly. You can see the revise manuscript.

2.3. It is better to revise line number 375-378 , It is not advisable to compare with USA studies , please find studies done in low income countries , Asian …..

We have accepted the comment and we modified it.

Finally it is better to revise spelling, grammar and vocabulary corrections on the whole section of a manuscript.

We have tried to fix the grammar and spelling error in the revised manuscript. 

Reviewers' comments:

Reviewer #1: Dear authors,

Thank you for your contribution to the field. I have had the opportunity to review your manuscript and found it well-written. However, there are some points you need to address to improve the quality of your work, and I have mentioned them section by section as follows:

Title and Abstract

1. The title is interesting and it addresses the current public health problem, but you should rewrite it as “Suicidal ideation and its associated factor among pregnant women in the eastern part of Ethiopia: A cross-sectional study.”

Authors’ response: Dear reviewer, we have accepted the comment and rewrite the title as “Suicidal ideation and associated factors among pregnant women attending antenatal care at public hospitals of Harari regional state, eastern Ethiopia: A cross-sectional study” as suggested by you and editor. Thank you. 

2. In the methods section of the abstract, you should incorporate the specific study area and study period.

Authors’ response: We have accepted the comment and we included a specific study area and study period. Thank you. 

3.In the result section of the abstract, paraphrase the following sentence to make it easier for the reader: Experiencing intimate partner violence and stress (AOR= 3.46; at 95% CI= = 1.75–6.66) and (AOR = 2.88; at 95% CI = 1.11–7.36) were significantly associated with suicidal ideation among pregnant women. “

Authors’ response: We have accepted the comment and paraphrase it accordingly in the revised manuscript. Thank you. 

4. Put keywords in ascending order

Authors’ response: We have accepted the comment and we put the keywords in ascending order. Thank you. 

introduction 

5. Overall, the introduction is well-written and adheres to the systematic writing of the background (deductive approach), and it maintains coherence between paragraphs. However, try to improve the grammatical errors through proofreading.

Authors’ response: Dear reviewer, thank you for the comment. We have tried to improve the grammatical error in the revised version of the manuscript. 

Methods 

The methods are clear, and the statistical approaches used in this study are appropriate. I have only three suggestions to improve the paper:

6. The sample size calculation formula does not need to be included in the calculation. The author just needs to convey what formula is used along with the margin of error set to get the required sample size.

Authors’ response: We have accepted the comment and removed the formula as suggested. Thank you. 

7. In lines 102-107, you should specify your study area. In which hospital did you conduct your study? Let us know more about your study hospitals.

Authors’ response: We have accepted the comment and specified the study area including the two public hospitals where the study was done. Thank you. 

8. In lines 120–122, why do you exclude those with hyperemesis gravidarum secondary to other medical conditions such as thyroid diseases and liver diseases after proven investigation? It needs further justification if thyroid diseases and liver diseases have a direct linkage with suicidal ideation; if it is so, don’t forget to cite your reference for your justification.

Authors’ response: Thank you for the concern. One of the primary reasons for excluding individuals with hyperemesis gravidarum secondary to other medical conditions from the study was the complexity and heterogeneity of their medical condition. Thyroid diseases and liver diseases encompass a wide range of disorders, each with its own unique pathophysiology, clinical manifestations, and treatment approaches. Including individuals with hyperemesis gravidarum secondary to these diverse conditions in a study may introduce significant variability that could confound the results and make it challenging to draw meaningful conclusions. That is why we prefer to exclude them. Thank you. 

9. In line 127, why do you use a margin of error of 0.03 instead of 0.05?

Authors’ response: We took proportion (P) of 13.3% from the previous study conducted in Jimma as a magnitude of suicidal ideation among pregnant women, which was resulting small sample size. Therefore, in order to increase the sample size, the margin error of 3% was taken. Thank you. 

10.Line 135-140 Why do you take average monthly attendants since your study period is 2 months? You calculated the k value in the wrong way. Please readjust it. 

Authors’ response: In order to get the average flow of attendants, we dividing the annual total number into months. However, we made an error on the calculating the K value, we supposed to take the two month period, and we readjusted it in the revised manuscript as “The interval size (k) is calculated using the following formula. k=N/n =2448/543 = 4.5 ≈4.So every four persons was selected from each hospital”. Thank you for your constructive comment. 

11. In Line 141, why do you select from the first three since your k value is 2?

Authors’ response: We took the comment and corrected it in the revised version of the manuscript as “The first pregnant woman was selected from the first four by lottery method”. Thank you. 

12. In lines 143–146, you should provide the questionnaire that was used for your study in a supplementary file and cite it in the method section as a supplemental file.

Authors’ response: We have accepted the comment and we have uploaded all the data underlying the findings described in the manuscript as supplementary information and we have cited it as supplemental file in data collection tool sub-section in the methods section of the revised the manuscript. Thank you. 

13. In Line 166 (IPV), you should provide both full words and abbreviations when you use it for the first time.

Authors’ response: We have accepted and addressed the comment. Thank you. 

14. In lines 168–169, move the sentence and merge it under data quality control.

Authors’ response: We have accepted and addressed the comment. Thank you.

15. In data quality control, did you provide training for data collectors? If so, you should state it in your manuscript.

Authors’ response: Yes we provide the training for data collectors regarding data collection methods, data collection tools, and how to handle ethical issues. We have stated this in the revised manuscript. Thank you. 

16. In Line 170-189, move your operational definitions above the data collection instruments and procedures.

Authors’ response: We have accepted and addressed the comment. Thank you. 

17. Cite reference for the operational definitions of pregnant women, hyperemesis gravidarum, Suicidal ideation and sleep quality

Authors’ response: We have accepted and addressed the comment. Thank you.

Results

The results are relevant, but you should revise them based on the following comment to improve the manuscript.

18.Line 214, “{26.53” should be replaced by "26.53 years.”

19. Line 215, do you think the word most is suitable for 58.36%?

20. Line 217, delete “with the largest proportion."

Authors’ response: We have accepted and addressed the comments from 18 to 20. Thank you.

21. In Table 1, you should re-categorize the age by using the standard age category. If you have a study participant whose age is less than 18, you should specify it by adding the category <18 years, since it will provide us with good results.

Authors’ response: Thank you for your concern. We have got only two participants who aged than 18 years and this small number makes us unable to compare them with other age category. That is why we have merged it as less than 20 years. 

22. It is difficult to assess the average monthly income for your study participants since either they are not civil servants or unless you did a wealth index.

Authors’ response: We agree with your concern. Since we didn’t do the wealth index and less reliable to put it in income, we omit it from the revised manuscript. Thank you. 

23. Please try to change Table 4 into a pie chart format

Authors’ response: We have accepted the comment and changed table 4 into pie chart. Thank you. 

24. Lines 260–262, rewrite each factor separately: “ The odds ratio of having suicidal ideation for a pregnant mother with depressive and anxiety symptoms were 2.79 and 3.37 compared to their counterparts (AOR = 2.79; (95% CI= = 1.49–5.23) and AOR = 3.37; (95% CI = 1.69–6.68), respectively.”

Authors’ response: We have accepted the comment and we rewrite it as suggested. Thank you. 

25. In Table 5, age less than 20, how do you run the regression for cell value less than 5, which is "3"? It is a critical issue and must be addressed by re-categorizing the age value.

Authors’ response: Thank you for your concern. The regression can run the cell value if it is not 0, but the total frequency of the specific category should be more than 5, which is 27 in this case. That is why we categorize it as mentioned in the manuscript. Thank you. 

26. In the Table 5 suicidal ideation column, put the suicidal value of yes before no for a better understanding

Authors’ response: We have accepted and addressed the comment as suggested. Thank you.

Discussion 

The discussions are explained well enough and based on the results. But to improve it, you should revise it based on the following comments:

27. In Line 282, try to justify how the tool difference is the possible reason for the variation. Try to justify by raising ideas about each tool and their possibility to increase or decrease their estimation of prevalence. Please try to think beyond tool difference and inclusion criteria

28. Lines 290–292, try to justify how the tool difference is the possible reason for the variation. Try to justify by raising ideas about each tool and their possibility to increase or decrease their estimation of prevalence. Please try to think beyond tool difference and inclusion criteria

Authors’ response: We have accepted the comment and tried to describe the possible explanations for the discrepancy occurred between the current study and previous studies other than tool difference in the revised manuscript. Additionally, we also elaborated how the tools difference affects the outcome (suicide). Thank you.

29. Line 301, you should specify by stating the specific study area for “This is supported by studies (13–15).”

Authors’ response: We have accepted and addressed the comment as suggested. Thank you.

30. Line 301-303 is not directly linked to your justification. You should better delete it. “A cross-sectional study conducted in the United Kingdom reported that 52.1 percent of participants thought to terminate their pregnancy, and 4.9 percent of them terminated their pregnancy owing to hyperemesis gravidarum.”

Authors’ response: We have accepted the comment and deleted the stated statement since it is not a relevant justification. Thank you.

Strengths and Limitations of the study

31. Line 344, please clarify this sentence:evidence-based laboratory tests Which lab? To assess what?

32. In lines 344–345, it is good to state recall bias and social desirability bias as limitations, but you should better state your effort to reduce those biases in this section.

33.Line 344–346 is a non-sense paragraph; you are expected to paraphrase it “However, recall bias, social desirability, a cross-sectional study design that cannot allow establishing a temporal relationship between suicidal ideation, and significant associated factors were the limitations of the study.

Authors’ response: For question 31 to 33, we have accepted the comments and we extensively rewrite the strength and limitations 

section in the revised version of the manuscript. 

Conclusion

34. Your conclusion is aligned with the implication of your study rather than the mere figure of the results and is also drawn from your main finding. I am okay with that.

35. In the recommendation section, why do you only stick with hyperemesis gravidarum since you have many findings?

Authors’ response: We have accepted the comment and we added the recommendations for the remaining main findings. Thank you.

Reviewer #2: 1.It will be more helpful to the readers if the setting where the study was conducted is mentioned in the abstract part.

Authors’ response: We have accepted the comment and we included a specific setting were the study was conducted in the revised version of the manuscript. Thank you. 

2.It also makes the research more influential if the research owner gives full feedback on the findings. For example, unwanted pregnancy is a factor for SI, but there is nothing to what should be done

Authors’ response: We have accepted the comment and we added the recommendations for the all main findings as suggested. Thank you.

3.It would not be good to use one or two of the references that are out of date

Authors’ response: We have accepted the comment and we have tried to use latest references. Thank you. 

4. The Overview of suicidal ideation among the study population is not organized well based on Global to local contexts. .

Authors’ response: We have accepted the comment and re-arranged the introduction part as suggested. Thank you. 

5.author text document spacing should be double spacing according to the journal guide line

Authors’ response: We have checked the journal guideline and made the adjustments to meet the journal requirements. Thank you. 

---

## [Editor Report · Decision Letter 2]

27 Feb 2024

Suicidal ideation and associated factors among pregnant women attending antenatal care at public hospitals of Harari regional state, Eastern Ethiopia: A cross-sectional study

PONE-D-23-31139R2

Dear Dr. Gebremariam,

We’re pleased to inform you that your manuscript has been judged scientifically suitable for publication and will be formally accepted for publication once it meets all outstanding technical requirements.

Kind regards,

Chalachew Kassaw Demoze, Msc

Academic Editor

PLOS ONE

Additional Editor Comments (optional):

Dear author , please revise the following issues:

1. " Various studies have reported that hyperemesis gravidarum can be associated with suicide (14–16). Hyperemesis gravidarum is also one of the predisposing factors for depression, anxiety, and psychological problems, (14-16). "It is better to use a single reference for both .

2. it is better to re-write a discussion citation on both controversial issues "Brazil, 12.55% (72), is included both on inline and lower than the finding." , so it is better to revise it

3. It is better to revise again about the selection of studies done in the USA for discussion , almost you used 4 USA studies for discussion (references 16, 74, 83, 82). It is better to find studies done from comparable socio-economic status, such as in Asia, Latin America, and Africa.

4. It is better to revise grammar, spelling, and consistency throughout the whole body of the manuscript.
---

## [Editor Report · Acceptance letter]

5 Mar 2024

PONE-D-23-31139R2 

PLOS ONE

Dear Dr. Gebremariam, 

I'm pleased to inform you that your manuscript has been deemed suitable for publication in PLOS ONE. Congratulations! Your manuscript is now being handed over to our production team.

Kind regards, 

on behalf of

Mr Chalachew Kassaw Demoze 

Academic Editor

PLOS ONE